# Nano-Conversion of Ineffective Cephalosporins into Potent One against Resistant Clinical Uro-Pathogens via Gold Nanoparticles

**DOI:** 10.3390/nano13030475

**Published:** 2023-01-24

**Authors:** Syed Mohd Danish Rizvi, Talib Hussain, Farhan Alshammari, Hana Sonbol, Nabeel Ahmad, Syed Shah Mohammed Faiyaz, Mohammad Amjad Kamal, El-Sayed Khafagy, Afrasim Moin, Amr Selim Abu Lila

**Affiliations:** 1Department of Pharmaceutics, College of Pharmacy, University of Ha’il, Ha’il 81442, Saudi Arabia; 2Molecular Diagnostic & Personalized Therapeutic Unit, University of Ha’il, Ha’il 81442, Saudi Arabia; 3Department of Pharmacology and Toxicology, College of Pharmacy, University of Ha’il, Ha’il 81442, Saudi Arabia; 4Department of Biology, College of Science, Princess Nourah bint Abdulrahman University, Riyadh 11671, Saudi Arabia; 5Department of Biotechnology, School of Allied Sciences, Dev Bhoomi Uttarakhand University, Dehradun 248007, India; 6Department of Physiology, College of Medicine, University of Ha’il, Ha’il 81442, Saudi Arabia; 7Institutes for Systems Genetics, Frontiers Science Center for Disease-Related Molecular Network, West China Hospital, Sichuan University, Chengdu 610065, China; 8King Fahd Medical Research Center, King Abdulaziz University, Jeddah 21589, Saudi Arabia; 9Department of Pharmacy, Faculty of Allied Health Sciences, Daffodil International University, Dhaka 1207, Bangladesh; 10Novel Global Community Educational Foundation, Hebersham, NSW 2770, Australia; 11Department of Pharmaceutics, College of Pharmacy, Prince Sattam Bin Abdulaziz University, Al-Kharj 11942, Saudi Arabia; 12Department of Pharmaceutics and Industrial Pharmacy, Faculty of Pharmacy, Suez Canal University, Ismailia 41522, Egypt

**Keywords:** antibiotic resistance, cefotetan, cefixime, clinical pathogens, gold nanoparticles

## Abstract

Infections caused by resistant bacterial pathogens have increased the complications of clinicians worldwide. The quest for effective antibacterial agents against resistant pathogens has prompted researchers to develop new classes of antibiotics. Unfortunately, pathogens have acted more smartly by developing resistance to even the newest class of antibiotics with time. The culture sensitivity analysis of the clinical samples revealed that pathogens are gaining resistance toward the new generations of cephalosporins at a very fast rate globally. The current study developed gold nanoparticles (AuNPs) that could efficiently deliver the 2nd (cefotetan-CT) and 3rd (cefixime-CX) generation cephalosporins to resistant clinical pathogens. In fact, both CT and CX were used to reduce and stabilize AuNPs by applying a one-pot synthesis approach, and their characterization was performed via spectrophotometry, dynamic light scattering and electron microscopy. Moreover, the synthesized AuNPs were tested against uro-pathogenic resistant clinical strains of *Escherichia coli* and *Klebsiella pneumoniae*. CT-AuNPs characteristic SPR peak was observed at 542 nm, and CX-AuNPs showed the same at 522 nm. The stability measurement showed ζ potential as −24.9 mV and −25.2 mV for CT-AuNPs and CX-AuNPs, respectively. Scanning electron microscopy revealed the spherical shape of both the AuNPs, whereas, the size by transmission electron microscopy for CT-AuNPs and CX-AuNPs were estimated to be 45 ± 19 nm and 35 ± 17 nm, respectively. Importantly, once loaded onto AuNPs, both the cephalosporin antibiotics become extremely potent against the resistant strains of *E. coli* and *K. pneumoniae* with MIC_50_ in the range of 0.5 to 0.8 μg/mL. The findings propose that old-generation unresponsive antibiotics could be revived into potent nano-antibiotics via AuNPs. Thus, investing efforts, intellect, time and funds for a nano-antibiotic strategy might be a better approach to overcome resistance than investing the same in the development of newer antibiotic molecule(s).

## 1. Introduction

The global prevalence of infections caused by antibiotic-resistant bacterial pathogens has markedly increased in the past few decades [1]. There is a plethora of reports that suggested “antibiotic resistance” as an impending threat to the human population [2,3,4]. In addition, WHO has listed resistance in microorganisms as one of the ten most serious health threat to the global population [5]. Despite continuous warnings, the inappropriate use of antibiotics and self-medication has not stopped. On the other hand, bacterial pathogens have acted smartly by developing different types of resistance mechanisms to evade the action of conventional antibiotics [6,7,8,9,10]. Thus, the current grave situation demands better alternative therapeutic options. Interestingly, nanoparticles could deliver antibiotics effectively to the resistant pathogen, and aid in converting unresponsive antibiotics into potent ones [11,12,13,14,15,16,17,18,19]. In fact, a large surface area: volume ratio of nanoparticles enables a number of antibiotic molecules to attach to it and prepare a multivalent nano-antibiotic against the resistant pathogen [20]. Further, nanoparticles themselves possess antibacterial potential via the generation of ROS and interaction with bacterial biomolecules (DNA, RNA, protein, enzymes), cell walls, and cell membranes. Although the mechanism of antibacterial action of nanoparticles is not fixed, this can be considered as a blessing in disguise to tackle resistance aspects of bacterial pathogens. 

Nanoparticle that has gained the limelight during the past decade due to their exceptional features is gold nanoparticles (AuNPs). AuNPs are not only receiving attention from researchers all over the world, but big companies such as Sigma-Aldrich, NanoHybrids, Cytodiagnostics, Goldsol, etc. are also investing in them [21]. The worldwide market for AuNPs was approximately four and a half billion US dollars in 2021 which is expected to be eight billion US dollars by 2027 [22,23]. Importantly, AuNPs have been widely explored for their potential against antibiotic-resistant strains of different pathogens. In fact, AuNPs have helped to form successful nano-antibiotics by effectively grafting antibiotic molecules on their surface without compromising or damaging the active moiety of the attached antibiotic [19,20,24,25]. Moreover, the synthesis of AuNPs-based nano-antibiotics does not require a complex technique, a simple one-pot synthesis approach where the antibiotic itself acts as a reducing and capping agent could be adequate [11,12,13,14,15,24]. In the present study, AuNPs were explored as a delivery tool for two different generations of cephalosporin antibiotics i.e., 2nd generation- cefotetan (CT) and 3rd generation-cefixime (CX). The successive generations (1st to 5th) of cephalosporins have been developed to increase their potency against different bacterial pathogens (with a prime focus on gram-negative bacteria). However, bacterial pathogens have gained resistance to almost all the generations of cephalosporins with time. Thus, converting unresponsive old-generation cephalosporin antibiotics into effective nano-antibiotics via AuNPs has its due clinical relevance. 

The present study successfully developed nano-antibiotics of CT and CX and compared their antibacterial potential against resistant gram-negative uro-pathogenic strains. It is noteworthy to mention that *Escherichia coli* and *Klebsiella pneumoniae* clinical strains used in the present study were not only resistant to cephalosporins but showed resistance towards β-lactamase inhibitor combination and other classes of antibiotics. However, prior to the exploration of antibacterial potential, both the nano-antibiotics (CT-AuNPs and CX-AuNPs) were duly characterized by spectrophotometry, dynamic light scattering and electron microscopy. In addition, loading efficiency was also calculated for both the nano-formulations and taken into consideration while calculating their antibacterial strength or MIC_50_. 

## 2. Materials and Methods

### 2.1. Materials

Cefotetan (CT), cefixime (CX), gold chloride salt, culture media, chemicals and solvents were procured from Sigma Aldrich (St. Louis, MO, USA).

### 2.2. AuNPs Synthesis

Second-generation (CT) and third-generation (CX) cephalosporin antibiotics at 250 μg concentration were added to a 3 mL reaction mixture containing 1 mM gold chloride salt (prepared in 7.2 pH phosphate buffer), separately [11]. Further, both the reaction mixtures were incubated at 40 °C for two days. The synthesis of AuNPs was visually confirmed by the color change i.e., from pale yellow to ruby red. Centrifugation at 30,000× g was performed for 30 min to collect synthesized AuNPs. Centrifuged AuNPs were further washed with milli-Q water and ethanol.

### 2.3. AuNPs Characterization

Characterization of CT and CX synthesized AuNPs were performed by spectrophotometry, zeta sizing and potential, and electron microscopy.

#### 2.3.1. Spectrophotometry

UV-Visible double-beam spectrophotometer (UV-1601, Shimadzu, Tokyo, Japan) was used to scan (from 200 nm to 800 nm) the transformation of gold salt into AuNPs at 1 nm resolution. 

#### 2.3.2. Zeta Sizing and Zeta Potential

The synthesized CT-AuNPs and CX-AuNPs were sonicated (1 min) and filtered (0.45 μm membrane filters) before analyzing them on Malvern Nano Zetasizer (ZEN3600, Malvern Instrument Ltd., Malvern, UK). The mean hydrodynamic diameter of both the AuNPs was calculated by using DTS0112 cuvette, whereas the surface zeta potential for each AuNPs was estimated by using DTS1070 cuvette [16].

#### 2.3.3. Electron Microscopy

Scanning electron microscopy (SEM) and Transmission electron microscopy (TEM) analysis was performed for both CT-AuNPs and CX-AuNPs.

FEI quanta 250 SEM (FEI Company, Hillsboro, OR, USA) at 30 kV acceleration voltage was used to collect SEM images of the synthesized AuNPs. Prior to SEM analysis, samples of CT-AuNPs and CX-AuNPs were deposited on the surface of a silicon substrate (conductive) and dried by using a hotplate at 60 °C.

TEM (Tecnai G2 Spirit) with a fitted BioTwin lens (Hillsboro, OR, USA) at 80 kV accelerating voltage was used to collect TEM images. However, the CT-AuNPs and CX-AuNPs samples were fixed on a carbon-coated copper grid before TEM analysis. 

### 2.4. Estimation of Loading Efficiency

The loading efficiency of CT and CX onto AuNPs was calculated with the help of UV-Vis spectrophotometer as described by Alshammari et al. [15]. After AuNPs synthesis by CT and CX, the 3 mL reaction mixtures were centrifuged for 30 min at 30,000× g and the supernatants were cautiously taken into a separate falcon tube. The concentration of unbound CT and CX were estimated in the supernatant at 254 nm [26] and 290 nm [27], respectively by using their pre-determined calibration curve. Further, the loading efficiency % for each AuNPs sample was estimated by applying the following formula [28]:% of loading efficiency = [(Y − Z)/Y] × 100(1)
where, Y is the initial concentration of CT or CX used for the AuNPs synthesis, and Z is the remaining (unbound) concentration of CT or CX in the supernatant. 

### 2.5. Assessment of Antibacterial Potential of CT-AuNPs and CX-AuNPs

#### 2.5.1. Resistant Uropathogenic Strains

Uro-pathogenic resistant strains of *Klebsiella pneumoniae* (Seq# 427811998026) and *Escherichia coli* (Seq# 427812372404) were obtained from Hail General Hospital, Hail, Saudi Arabia. Both the pathogenic strains were inoculated in a fresh nutrient broth medium and incubated at 37 °C for 18 h. Prior to antibacterial evaluation, the turbidity for each strain was maintained up to 1.5 × 108 CFU/mL (0.5 McFarland standard).

#### 2.5.2. Antibacterial Assessment by Agar Well Diffusion

Preliminary antibacterial assessment of CT-AuNPs and CX-AuNPs was performed by agar well diffusion method [29]. Mueller–Hinton agar plates were swabbed with a fresh inoculum of each uro-pathogenic strain. After that well cutter was used to aseptically punch four holes (two holes of 4 mm and two holes of 8 mm) on the swabbed agar plates. Pure CT (without AuNPs) and CT-AuNPs were added to the wells at 3.25 μg/well and 6.5 μg/well. Similar concentrations were applied for pure CX and CX-AuNPs. All the Petri plates were incubated for 18 h at 37 °C, and the zone of inhibition was measured. The final zone of inhibition in mm is a mean of triplicate experiments.

#### 2.5.3. Antibacterial Assessment by Calculating Minimal Inhibitory Concentration

The MIC_50_ of pure antibiotic (CT and CX) and its gold nanoformulation (CT-AuNPs and CX-AuNPs) against uropathogenic strains of *K. pneumoniae* and *E. coli* were estimated by using microbroth dilution technique [30]. The concentration of both pure antibiotic and antibiotic after loading to AuNPs was kept in a range from 0.126 to 65 μg/mL in 96-well microtiter plates. After concentration adjustment, 10 μL of test strain (at 0.5 McFarland standard) was added to each well. All the microtiter plates were incubated for 18 h at 37 °C, and MIC_50_ was calculated. The minimum concentration at which the growth was inhibited was noted as MIC, however, triplicate experiments were performed to calculate the mean ± standard deviation of MIC values.

## 3. Results and Discussion

New-generation cephalosporins have been developed from time to time with an aim to provide better antibiotic therapy for the human population [31]. However, bacterial pathogens have gained resistance against almost all generations of cephalosporin [32,33,34]. Pathogens have developed different resistance mechanisms such as antibiotic efflux, decrease in antibiotic uptake, disarming antibiotics and manipulating antibiotic targets to escape the cidal/static effect of antibiotics [10]. All these mechanisms could work for developing cephalosporin resistance but particularly disarming by β-lactamase enzyme is considered the major resistance mechanism. Hence, to combat β-lactamase enzyme, β-lactamase inhibitors were added as an adjuvant with the antibiotics. It showed a remarkable effect and provided a strong hope against resistant bacterial pathogens [35]. Unfortunately, pathogens were smart enough to develop resistance towards even the combination of antibiotic and β-lactamase inhibitors [36,37]. Thus, scientists are working hard to explore alternative therapeutic options against resistant bacterial pathogens. Here, in the present study, gold nanoformulations of old-generation (2nd and 3rd) cephalosporins were developed and tested against resistant strains of *E. coli* and *K. pneumoniae*.

### 3.1. Synthesis of CT-AuNPs and CX-AuNPs

AuNPs of various sizes and features are synthesized *via* chemical reducing agents or natural extracts/enzymes (green synthesis) [38,39,40,41,42]. In chemical synthesis approaches, chemicals such as sodium borohydrate, trisodium citrate, and hydrazine are used as reducing agents; however, AuNPs synthesized after chemical reduction sometimes need additional capping agents for stabilization [43]. On the other hand, plant extracts, micro-organisms and natural enzymes are applied as alternatives to chemicals for green synthesis of AuNPs [44,45]. However, the exact reducing and capping agent is difficult to decipher from a plethora of compounds from the natural extract. Moreover, loading/attachment of desired drug onto AuNPs is itself a tedious job that might include the use of harsh chemicals such as 1-ethyl-3-(3-dimethylaminopropyl) carbodiimide [19,46]. Hence, researchers developed one pot-AuNPs synthesis approach where the drug itself acts as a reducing and stabilizing agent to lessen the use of chemicals and ease the process of drug loading [11,12,13,14,15,24]. A similar approach was applied in the current study to synthesize AuNPs by two different generations of cephalosporins [2nd generation-Cefotetan (CT) and 3rd generation-Cefixime (CX)]. Both the cephalosporin antibiotics reduced and stabilized AuNPs in a single step (Figure 1). It is noteworthy to mention that the amine group of β-lactam antibiotics (cephalosporins) usually contributes to gold/silver salts reduction, and the β-lactam ring remains intact for its activity on the surface of nanoparticles even after stabilization or capping [20,24,25,47].

To synthesize AuNPs, CT and CX were added at 250 μg concentration to the reaction mixture containing gold salt in pH 7.2 buffer. It has been reported that pH could influence AuNPs zeta potential and stability [48]. In fact, a recent study [49] has shown the influence of pH change on the size and zeta potential of AuNPs. The authors showed that a pH value from 6 to 10 was most appropriate for the synthesis of stable AuNPs. They also observed that acidic pH was unfavorable for AuNPs synthesis with almost neutral zeta potential, large size and aggregation. However, in the current study, the pH was kept at 7.2 (physiological pH) on the basis of earlier reports [11,12,13,14,15,16,42]. The color change from pale yellow to ruby red (Figure 1) visually confirmed the synthesis of CT- and CX-AuNPs. Both CT-AuNPs and CX-AuNPs were further characterized via spectrophotometry, zeta sizer and electron microscopy.

### 3.2. Characterization of CT-AuNPs and CX-AuNPs

The first characterization of synthesized AuNPs was performed by UV-Vis spectrophotometer on the basis of characteristic surface plasma resonance (SPR) of AuNPs (i.e., in between the visible range of 500 nm to 600 nm). SPR band peaks for CT-AuNPs and CX-AuNPS were observed at 542 nm and 522 nm, respectively (Figure 2). Additional peaks at 254 nm [26] and 290 nm [27] were also observed during spectrophotometric scanning that corresponds to CT and CX, respectively, which suggested the successful loading of CT and CX onto AuNPs (Figure 2). In accordance, similar dual peaks have been reported by several recent investigations on antibiotic-mediated AuNPs synthesis [11,12,13,14,15]. The AuNPs synthesized by cefoxitin, delafloxacin, cefotaxime, vancomycin and ceftriaxone showed a second peak at 235, 290, 260, 278 and 241 nm, respectively, along with the characteristic AuNPs peak between 500 to 600 nm. Even the conjugation of antibiotics onto AuNPs with chemical agent EDC (instead of a one-step synthesis approach) also showed dual peaks that confirm the successful loading of the antibiotics to AuNPs [19,46].

The zeta size distribution of CX-AuNPs and CT-AuNPs in the dispersion was estimated by a dynamic light scattering approach. The size was estimated by measuring the arbitrary variations of the scattered intensity of light due to AuNPs dispersion [50]. The Z-average mean size for CX-AuNPs and CT-AuNPs was 117 nm and 119.6 nm, respectively (Figure 3). In addition, the colloidal stability of CX-AuNPs and CT-AuNPs was measured by zeta potential that estimates the surface charge on AuNPs. In fact, it depends on the electrostatic repulsion strength of the similarly charged particles in the dispersion [51]. The Zeta potential of CX-AuNPs and CT-AuNPs was estimated as –25.2 mV and –24.9 mV, respectively. The higher positive or negative zeta potential i.e., more than ±20 mV, repels the particles and minimizes the probability of aggregation [52]. Thus, CX-AuNPs and CT-AuNPs synthesized in the current study showed long-term stability even at room temperature for months. However, the functional group present on the antibiotics (CX and CT) might have attributed to the negative charge on the surface of AuNPs.

Scanning Electron Microscopy (SEM) scans the nanomaterial surface with a focused beam of high-energy electrons. The electron beam interacts with the nanomaterial to create signals that could be further translated into information pertinent to the surface morphology of the nanomaterial [53]. Here, in the present investigation, SEM was used to observe the surface morphology of the synthesized CX-AuNPs and CT-AuNPs, and both the AuNPs appeared spherical in shape (Figure 4). These findings were in accordance with the previous investigations, where antibiotics were loaded/conjugated to AuNPs and a similar spherical shape pattern was observed [15,19,46].

To estimate the size of inorganic core of CX-AuNPs and CT-AuNPs, transmission electron microscopy (TEM) was applied. TEM images were taken at 1,000,000× magnification for both the AuNPs (Figure 5). TEM analysis revealed that both AuNPs samples were poly-dispersed, with mean size of 35 ± 17 nm and 45 ± 19 nm for CX-AuNPs and CT-AuNPs, respectively. There was no observed aggregation in the TEM and SEM images that corresponds to the successful capping of AuNPs by both the antibiotics (CX and CT). It is a fact that different measurement techniques will show variation in size of nanoparticles. Dynamic light scattering approach calculate the size including the adhered solvent layer, while TEM estimates the size of inorganic core [54,55]. Hence, size determination by TEM is comparatively lower than size by DLS. Similar aspects of size differences have been observed in various previous reports on antibiotic loaded AuNPs [11,12,13,14,15,19].

Furthermore, the loading efficiency of CX-AuNPs and CT-AuNPs was estimated by using the formula described by Gomes et al. [28]. In fact, the calculation of loading efficiency is a crucial factor for the nanoformulation characterization and their biological application. The loading efficiency of CX and CT on AuNPs was calculated as 77.48% and 79.84%, respectively. Initially, 250 μg of both antibiotics (CX and CT) were added to the reaction mixture. Out of 250 μg, 193.7 μg of CX was loaded onto CX-AuNPs and 199.6 μg of CT was loaded onto CT-AuNPs. There was no significant loss of antibiotics observed during the synthesis of AuNPs, and an appropriate amount has been successfully loaded onto AuNPs. 

### 3.3. Comparative Antibacterial Assessment of CT-AuNPs and CX-AuNPs

In the past, several prevalence studies have confirmed the rapid expansion of cephalosporin resistance among gram-negative bacterial pathogens, particularly in clinical strains of *K. pneumoniae* and *E. coli* [56,57,58,59,60]. The present study designed nanoformulations of old-generation (second and third) cephalosporins against cephalosporin-resistant clinical strains of *K. pneumoniae* and *E. coli*. Comparative antibacterial assessment of AuNPs loaded with second generation-cefotetan (CT) and third-generation-cefixime (CX) was performed initially by well-diffusion technique (Figure 6 and Table 1) followed by MIC_50_ calculation (Figure 7). At 3.25 mg and 6.5 mg concentrations of CT-AuNPs, the zone of inhibitions against *E. coli* were observed as 18mm and 23mm, respectively (Figure 6a). Whereas, inhibition zones against *K. pneumoniae* were 19 mm and 24 mm at the same concentrations of CT-AuNPs (Figure 6b). On the other hand, CX-AuNPs at 3.25 mg and 6.5 mg concentrations showed inhibition zones against *E. coli* as 17 mm and 22 mm, respectively (Figure 6c). However, the zone of inhibitions against *K. pneumoniae* of CX-AuNPs at the same concentrations was estimated as 18 mm and 23 mm, respectively (Figure 6d). It is noteworthy to mention that at 3.25 mg and 6.5 mg concentrations neither pure CT nor pure CX showed any activity against the tested strains (Figure 6). Thus, it could be inferred from the results that CT and CX become potent against the tested strain after loading onto AuNPs at 3.25 mg and 6.5 mg concentrations. Both the tested bacterial pathogens belong to *Enterobactericiae* family, and according to Clinical and Laboratory Standards Institute (CLSI 2020) guidelines, the breakpoint of CT (30 mg) for sensitivity is ≥16 mm zone of inhibition and for CX (5 mg) sensitivity breakpoint is ≥19mm against *Enterobacterales.* In the present study, cefotetan (after loading onto AuNPs) at very low concentration (3.25 mg) showed significant activity i.e., more than 16 mm inhibition zone (CLSI Breakpoint) against both *K. pneumoniae* and *E. coli*. Hence, it can be safely stated that a second-generation cephalosporin (CT) has become relatively more potent than third-generation cephalosporin (CX) after loading onto AuNPs. 

Agar well diffusion technique provided qualitative antibacterial assessment, and quantitative assessment was further performed by calculating the MIC_50_ concentration of antibiotic nanoformulations and pure antibiotics against the tested strains. A comparative analysis of MIC values of pure CT and CX, and their gold nanoformulations were performed on *E. coli* (Figure 7a) and *K. pneumoniae* (Figure 7b). MIC_50_ of CX-AuNPs and CT-AuNPs against *E. coli* were estimated as 0.8 mg/mL and 0.65 mg/mL, respectively; whereas, pure CX and CT showed MIC_50_ as 19mg/mL and 32.5 mg/mL, respectively. On the other hand, CX-AuNPs and CT-AuNPs showed MIC_50_ as 0.75 mg/mL and 0.5 mg/mL against *K. pneumoniae*; while, MIC_50_ of pure antibiotics, CX and CT were estimated as 17 mg/mL and 27 mg/mL, respectively. MIC sensitivity breakpoint for CT against *Enterobactericiae* is ≤16 mg/mL, however, for CX it is ≤1 mg/mL (CLSI 2020). MIC of both AuNPs comes well under the limit of sensitivity set by the CLSI guidelines. It is to be noted that there is a huge difference in MIC values of the pure antibiotics and their gold nanoformulations. MIC_50_ values for CX were decreased by 23.75 and 22.66 times when loaded onto AuNPs against *E. coli* and *K. pneumoniae*, respectively. In contrast, MIC_50_ values for CT were decreased by 50 and 54 times when loaded onto AuNPs against *E. coli* and *K. pneumoniae*, respectively. MIC findings were in accordance with the well-diffusion assay results that suggested CT showed more potency than CX when loaded with AuNPs against the tested strains.

Cephalosporin antibiotic(s) resistance in bacterial pathogens is generally associated with overexpression of extended-spectrum β-lactamase [6]. However, the close association of β-lactamase with the downregulation of porin channels and upregulation of efflux pump in bacterial pathogens significantly enhances their resistance towards cephalosporin [61,62,63]. In the present study, the bacterial strains tested were resistant to second and third-generation cephalosporins, and AuNPs convert them from ineffective antibiotics into effective antibiotic-nanoformulations. The enhanced antibacterial potential of cephalosporin-nanoformulations in the present investigation could be correlated with the ability of AuNPs to successfully deliver an ample amount of cephalosporins to resistant pathogens. It has to be noted that the active moiety i.e., β-lactam ring remains intact after conjugating/loading cephalosporins onto AuNPs, and due to the large surface-to-vol ratio significant amount of cephalosporin could be loaded on the AuNPs [20,24,25,47,64]. Thus, it could be proposed that β-lactamase might become saturated by the sufficient quantity of substrate (cephalosporin) received, meanwhile, the untouched cephalosporin could perform its usual action on the cell wall. In addition, AuNPs themselves have the ability to bind/inhibit the efflux pump, alter the permeability of the cell membrane and interact/disrupt the biomolecules of bacterial pathogens [17,24,65]. Therefore, it could be suggested that the antibacterial effect was due to the synergism of AuNPs and cephalosporins loaded onto them. Interestingly, AuNPs have no defined or single mechanism of action against bacterial pathogens, hence, developing resistance against them is quite a difficult task for the pathogens. The present investigation would like to emphasize one aspect: if an old generation cephalosporin could be resuscitated with AuNPs, “Why should the intellect, time and funds be spent on developing newer generations?”. AuNPs appear to be a smarter alternative to overcoming resistance to bacterial pathogens. However, cost-effectiveness and safe applicability of AuNPs are still a question of debate.

It is noteworthy to mention that our team has recently worked on two third-generation cephalosporins gold nanoformulations (ceftriaxone and cefotaxime) [13,15] and one second-generation cephalosporin gold nanoformulations (cefoxitin) [11], and found that second-generation cephalosporin-AuNPs was equally effective antibacterial than third-generation AuNPs. However, the bacterial strains tested in each of these studies were different. These findings prompted our team to find a comparative analysis of the same resistant clinical strains to get a better insight into the antibacterial action of two different generations loaded onto AuNPs. The results suggested that an ineffective old-generation cephalosporin (CT) could be converted into an effective nanoformulation and show better antibacterial potential than the new-generation cephalosporin (CX) nanoformulations. However, it is too early to come to any conclusion as the human toxicity part and the fate of these AuNPs are still a point of debate, and our team has started working on toxicity aspects and getting deeper insights into the mechanism of their antibacterial action. Nevertheless, it is strongly believed that the scientific community could provide safer nano-antibiotic against resistant pathogens in the near future. 

## 4. Conclusions

In the present study, the comparative analysis of second-generation cephalosporin (CT)-AuNPs and third-generation cephalosporin (CX)-AuNPs was performed against resistant clinical strains of *E. coli* and *K. pneumoniae*. A facile one-pot synthesis approach was used to successfully synthesize the CT-AuNPs and CX-AuNPs via using cephalosporins (CT and CX) as reducing and capping agents. The synthesized CT-AuNPs and CX-AuNPs were stable with ζ potential as −24.9 mV and −25.2 mV, and size as 45 ± 19 nm and 35 ± 17 nm, respectively. However, 79.84% of CT and 77.48% of CX were loaded onto the synthesized AuNPs. CT after loading to AuNPs becomes ~50 times more potent than the pure CT, while CX after loading to AuNPs becomes ~25 times more active than pure CX against the CT and CX-resistant tested strains. Hence, ineffective cephalosporins could be resuscitated into effective nano-antibiotic with the help of AuNPs. In addition, the idea of ‘nano-conversion of old generation cephalosporin’ appears to be more promising than spending efforts and intellects on developing a new generation of cephalosprorin. However, the toxicity and fate of AuNPs need to decipher in a planned manner. Nevertheless, the present investigation paved the way to design AuNPs-based nano-antibiotics to tackle resistance issues in bacterial pathogens.

## Figures and Tables

**Figure 1 nanomaterials-13-00475-f001:**
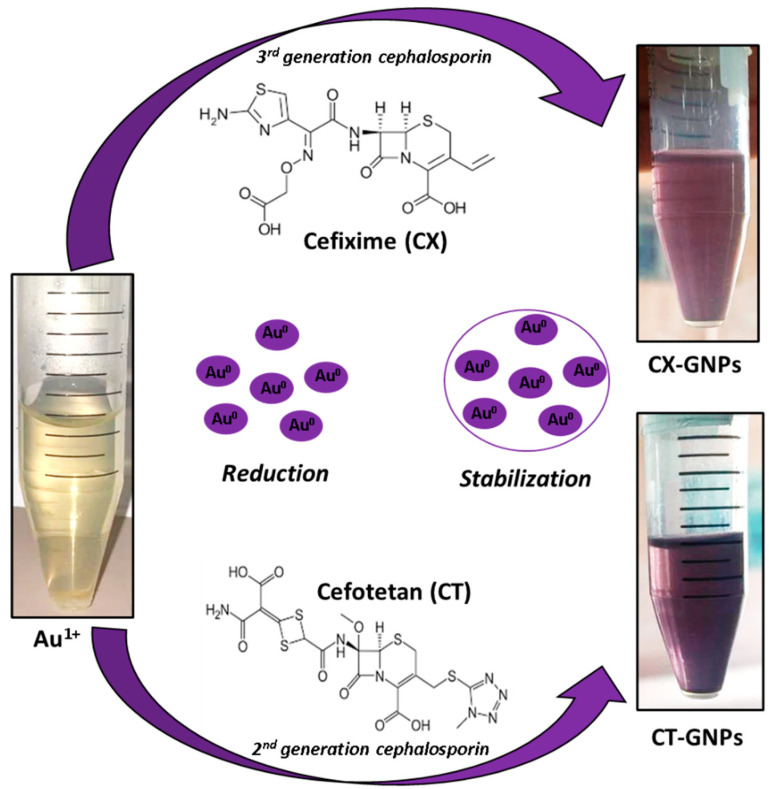
Schematic representation of AuNPs synthesis by cefotetan and cefixime.

**Figure 2 nanomaterials-13-00475-f002:**
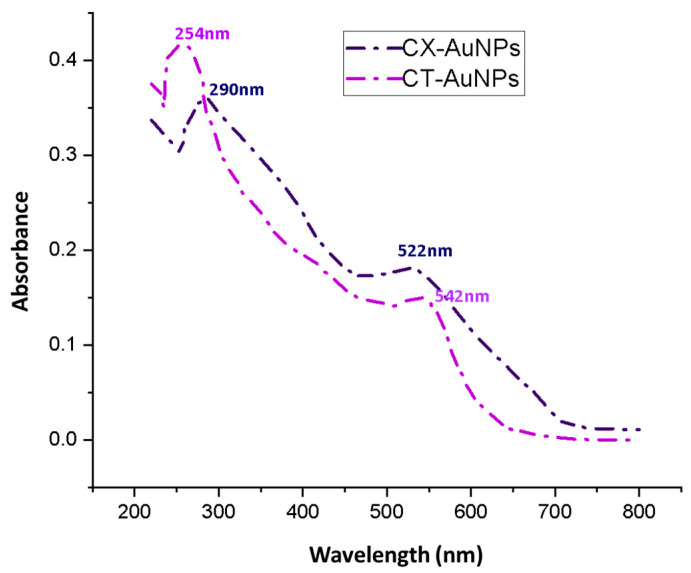
UV-Visible spectrophotometric scan of CX-AuNPs and CT-AuNPs.

**Figure 3 nanomaterials-13-00475-f003:**
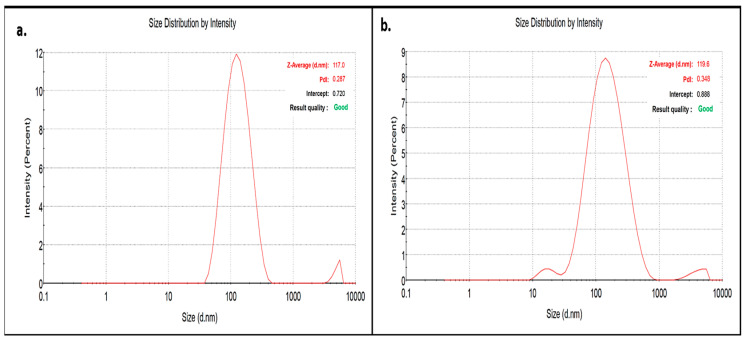
Size distribution vs intensity plot of (**a**) CX-AuNPs (**b**) CT-AuNPs.

**Figure 4 nanomaterials-13-00475-f004:**
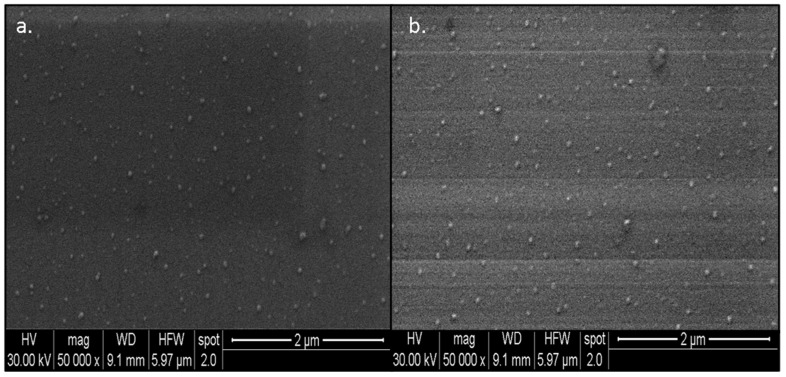
SEM images of (**a**) CX-AuNPs and (**b**) CT-AuNPs.

**Figure 5 nanomaterials-13-00475-f005:**
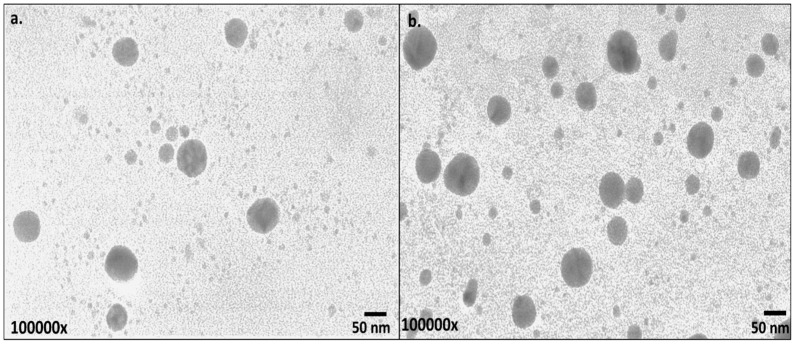
TEM images of (**a**) CX-AuNPs and (**b**) CT-AuNPs.

**Figure 6 nanomaterials-13-00475-f006:**
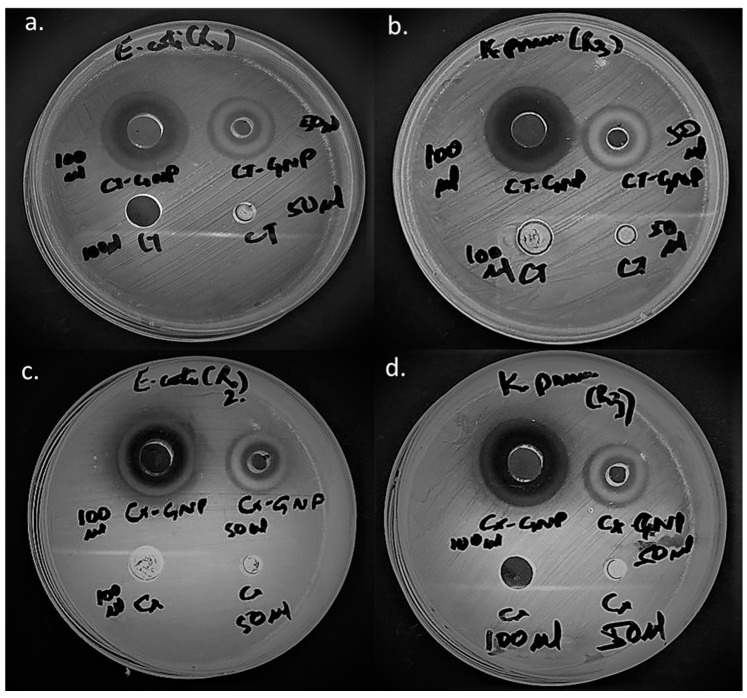
Antibacterial activity of (**a**) pure CT and CT-AuNPs against *E. coli*, (**b**) pure CT and CT-AuNPs against *K.pneumoniae*, (**c**) pure CX and CX-AuNPs against *E. coli*, and (**d**) pure CX and CX-AuNPs against *K.pneumoniae*.

**Figure 7 nanomaterials-13-00475-f007:**
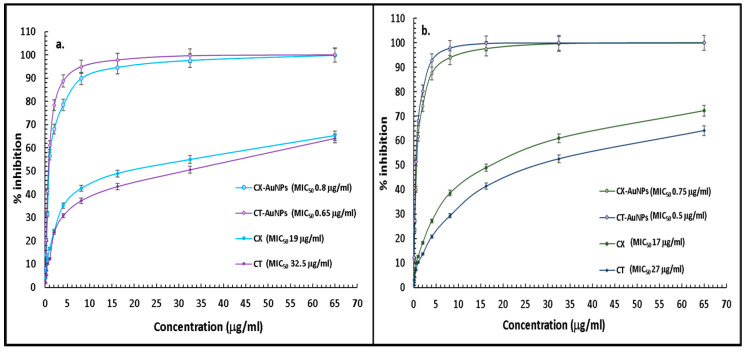
MIC graph of pure CT, pure CX, CT-AuNPs and CX-AuNPs against (**a**). *E. coli* and (**b**). *K. pneumoniae*.

**Table 1 nanomaterials-13-00475-t001:** : Inhibition zones of CT-AuNPs and CX-AuNPs against *E. coli* and *K. pneumoniae*.

	*E. coli*	*K. pneumoniae*
CX-GNPs (3.25 mg)	17 mm	18 mm
CX-GNPs (6.5 mg)	22 mm	23 mm
CT-GNPs (3.25 mg)	18 mm	19 mm
CT-GNPs (6.5 mg)	23 mm	24 mm

## Data Availability

Not applicable.

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
