# Peer review of "Nano-Conversion of Ineffective Cephalosporins into Potent One against Resistant Clinical Uro-Pathogens via Gold Nanoparticles"

_nanomaterials, 2023, doi:10.3390/nano13030475_

Round 1

Reviewer 1 Report

The under review manuscript is an important contribution in the scientific field of "nanomaterials". The presentation is distinct, and the results adequately documented. The literature is significant and informative for the scientists on the relevant fields. I suggest the publication of the submitted Journal. 

Author Response

Reviewer 1

The under-review manuscript is an important contribution in the scientific field of "nanomaterials". The presentation is distinct, and the results adequately documented. The literature is significant and informative for the scientists on the relevant fields. I suggest the publication of the submitted Journal. 

Reply: Thanks to the honorable reviewer for the appreciation of our efforts. English language has been duly revised in the MS.

Reviewer 2 Report

Gold nanoparticles show promising applications in the antibacterial field. In this manuscript, the authors successfully developed the nano-antibiotics CT and CX and compared their antibacterial potential against resistant gram-negative uro-pathogenic strains. The study is interesting, and the results are reliable. However, minor revisions are still required to address the following issues.

  1. A discussion of the main methods of obtaining AgNPs, the state of research in the field, and the main advantages and disadvantages of their use compared to the current research in the field
  2. Can you explain why pH 7.5 was chosen? I understood from the paper that, based on other studies... How does a higher or lower pH value affects the zeta potential and stability of nanoparticles (perhaps this should be added to the article).
  3. Figure 7 - Please pay attention to the fact that "CX-GNPs" and "CT-GNPs" should be revised as "CX-AuNPs" and "CT-AuNPs."
  4. Starting from the question, "Why should intellect, time, and funds be spent on developing newer generations?" AuNPs seem like a smarter way to get around bacterial pathogens' resistance, but how cost-effective and safe is this method in the long run? 

Author Response

Reviewer 2

Gold nanoparticles show promising applications in the antibacterial field. In this manuscript, the authors successfully developed the nano-antibiotics CT and CX and compared their antibacterial potential against resistant gram-negative uro-pathogenic strains. The study is interesting, and the results are reliable. However, minor revisions are still required to address the following issues.

  1. A discussion of the main methods of obtaining AgNPs, the state of research in the field, and the main advantages and disadvantages of their use compared to the current research in the field.

Reply: As per the suggestion of the honorable reviewer, the main methods of obtaining AuNPs have been duly added in the section 3.1 of result and discussion with focus on advantages and disadvantages compared to current research.

  1. Can you explain why pH 7.5 was chosen? I understood from the paper that, based on other studies... How does a higher or lower pH value affects the zeta potential and stability of nanoparticles (perhaps this should be added to the article).

Reply: As per the suggestion of the honorable reviewer, affects of pH on zeta potential and stability has been duly added in the section 3.1 of result and discussion.

  1. Figure 7 - Please pay attention to the fact that "CX-GNPs" and "CT-GNPs" should be revised as "CX-AuNPs" and "CT-AuNPs."

Reply: We sincerely apologize for the error. We have duly corrected it in the revised MS.

  1. Starting from the question, "Why should intellect, time, and funds be spent on developing newer generations?" AuNPs seem like a smarter way to get around bacterial pathogens' resistance, but how cost-effective and safe is this method in the long run? 

Reply: Honorable reviewer has pointed out a very valid point on cost-effectiveness and safely of AuNPs. We have duly added a sentence 'However, cost effectiveness and safe applicability of AuNPs is still a question of debate' in the end of the discussion. However, big companies are investing on AuNPs for different applications [1-3] and various clinical trials are going on to test AuNPs applicability [4]. Even, cost effective methods for AuNPs antibacterial application have been developed [5]. In addition, biocompatibility of AuNPs is dependent on the features of AuNPs and its fabrication technique [6-8]. Earlier, we have tested antibiotic conjugated AuNPs on human embryonic kidney cells-293 cells and does not find any toxicity [9]. These findings broadened the scope of medical applicability of AuNPs in near future.

  1. Allied market research, https://www.alliedmarketresearch.com/gold-nanoparticles-market-A08997
  2. Globenewswire. Available online: https://www.globenewswire.com/news-release/2022/10/13/2534150/0/en/Global-Gold-Nanoparticles-Market-to-Reach-7-6-Billion-by-2027.html
  3. IMARCGROUP. Available online: https://www.imarcgroup.com/gold-nanoparticles-market
  4. Zhang, R., Kiessling, F., Lammers, T. et al. Clinical translation of gold nanoparticles. Drug Deliv. and Transl. Res. 13, 378–385 (2023). https://doi.org/10.1007/s13346-022-01232-4
  5. Piktel, E., Suprewicz, Ł., Depciuch, J. et al. Varied-shaped gold nanoparticles with nanogram killing efficiency as potential antimicrobial surface coatings for the medical devices. Sci Rep 11, 12546 (2021). https://doi.org/10.1038/s41598-021-91847-3
  6. Arai, Y., Jee, S.Y., Kim, S.M. et al. Biomedical applications and safety issues of gold nanoparticles. Toxicol. Environ. Health Sci. 4, 1–8 (2012). https://doi.org/10.1007/s13530-012-0111-z
  7. Kus-Liśkiewicz M, Fickers P, Ben Tahar I. Biocompatibility and Cytotoxicity of Gold Nanoparticles: Recent Advances in Methodologies and Regulations. Int J Mol Sci. 2021 Oct 11;22(20):10952. doi: 10.3390/ijms222010952. PMID: 34681612; PMCID: PMC8536023.
  8. Kadhim RJ, Karsh EH, Taqi ZJ, Jabir MS. Biocompatibility of gold nanoparticles: In-vitro and In-vivo study. Materials Today: Proceedings. 2021 Jan 1;42:3041-5.
  9. Shaikh, S.; Rizvi, S.M.D.; Shakil, S.; Hussain, T.; Alshammari, T.M.; Ahmad, W.; Tabrez, S.; Al-Qahtani, M.H.; Abuzenadah, A.M. Synthesis and Characterization of Cefotaxime Conjugated Gold Nanoparticles and Their Use to Target Drug-Resistant CTX-M-Producing Bacterial Pathogens. J Cell Biochem 2017, 118, 2802-2808, doi:10.1002/jcb.25929.

Reviewer 3 Report

Although the results are interesting, a comparison of the  cephalosporine loaded gold nanoparticles with standard citrated ones, of the same size, would have been interesting to evidence the role of gold in the antibacterial effect and would have clarified the discussion. A simple assay on agar plate may suffice.

The size of the nanoparticle present a great heterogeneity and numerous small particles of less than 20 nm are present and it is for me doubtful that the given diameters have such a limited standard deviation. Small nanoparticles may have a better activity as they can reach more easily their target.

Author Response

Reviewer 3

  1. Although the results are interesting, a comparison of the cephalosporin loaded gold nanoparticles with standard citrated ones, of the same size, would have been interesting to evidence the role of gold in the antibacterial effect and would have clarified the discussion. A simple assay on agar plate may suffice.

Reply: We agree with the suggestion of the honorable reviewer, however, we would like to state few points that will hopefully clarify the doubts. First of all, the method applied in the current study does not use any chemical as a reducing agent. Here, only antibiotic was used to reduce and cap the AuNPs. Thus, in our opinion, comparative analysis will not be suitable if we use AuNPs synthesized via a different approach. It is noteworthy to mention that our team has been working on green synthesis of AuNPs of different sizes since 2014 [1,2] and tested them on different strains [3,4]. We observed no antibacterial potential of naked (without antibiotics) green synthesized AuNPs in our earlier reports [3,4]. Similarly, the resistant strains used in the present study did not show any susceptibility towards the green synthesized AuNPs of different sizes. The fabrication of AuNPs plays a crucial role in their biological activity. In addition, we have tested our antibiotic loaded AuNPs on human embryonic kidney cells-293 cells and did not find any toxicity. We are still working on deciphering the mechanistic aspect of their antibacterial action. Hopefully, in near future we could provide a better insight to the mechanism that will clarify the antibacterial role of gold in the synthesized AuNPs.

  1. Khan S., Rizvi S.M., Saeed M., Srivastava A.K., Khan M. A Novel Approach for the synthesis of gold nanoparticles using Trypsin. Adv. Sci. Lett. 2014;20:1061–1065. doi: 10.1166/asl.2014.5481.
  2. Khan, S.; Danish Rizvi, S.M.; Avaish, M.; Arshad, M.; Bagga, P.; Khan, M.S. A novel process for size controlled biosynthesis of gold nanoparticles using bromelain. Materials Letters 2015, 159, 373-376, doi:https://doi.org/10.1016/j.matlet.2015.06.118.
  3. Shaikh, S.; Rizvi, S.M.D.; Shakil, S.; Hussain, T.; Alshammari, T.M.; Ahmad, W.; Tabrez, S.; Al-Qahtani, M.H.; Abuzenadah, A.M. Synthesis and Characterization of Cefotaxime Conjugated Gold Nanoparticles and Their Use to Target Drug-Resistant CTX-M-Producing Bacterial Pathogens. J Cell Biochem 2017, 118, 2802-2808, doi:10.1002/jcb.25929.
  4. Alshammari, F.; Alshammari, B.; Moin, A.; Alamri, A.; Al Hagbani, T.; Alobaida, A.; Baker, A.; Khan, S.; Rizvi, S.M.D. Ceftriaxone Mediated Synthesized Gold Nanoparticles: A Nano-Therapeutic Tool to Target Bacterial Resistance. Pharmaceutics 2021, 13, doi:10.3390/pharmaceutics13111896.
  5. The size of the nanoparticle presents a great heterogeneity and numerous small particles of less than 20 nm are present and it is for me doubtful that the given diameters have such a limited standard deviation. Small nanoparticles may have a better activity as they can reach more easily their target.

Reply: Honorable reviewer has rightly pointed out that the size in TEM images were poly dispersed. We have reviewed and corrected the size in the revised version of the MS. We apologize for the error. It was indeed a polydisperse sample with small particles in the selected TEM image frame. We are in fact thankful to honorable reviewer to improve our MS.

Round 2

Reviewer 3 Report

I accept your detailed answers and thus the paper in its revised form.

A few minor corrections:

line 309 et 352 : Enterobacteriaceae (in lieu of Enterobactericiae)

lines 404-406 :Nevertheless, it is strongly believed that the scientific community could provide safer( and) nano-antibiotic  against resistant pathogens in the near future.

lines 417-418 : could be resuscinated resuscitated into

Author Response

A few minor corrections:

  1. line 309 et 352 : Enterobacteriaceae(in lieu of Enterobactericiae)

Reply: The correction has been made in the revised MS as suggested by the honorable reviewer.

  1. lines 404-406 :Nevertheless, it is strongly believed that the scientific community could provide safer( and) nano-antibiotic against resistant pathogens in the near future.

Reply: The correction has been made in the revised MS as suggested by the honorable reviewer.

  1. lines 417-418 : could be resuscinated resuscitatedinto

Reply: The correction has been made in the revised MS as suggested by the honorable reviewer.